# Trimethylamine-*N*-Oxide Postprandial Response in Plasma and Urine Is Lower After Fermented Compared to Non-Fermented Dairy Consumption in Healthy Adults

**DOI:** 10.3390/nu12010234

**Published:** 2020-01-16

**Authors:** Kathryn J. Burton, Ralf Krüger, Valentin Scherz, Linda H. Münger, Gianfranco Picone, Nathalie Vionnet, Claire Bertelli, Gilbert Greub, Francesco Capozzi, Guy Vergères

**Affiliations:** 1Federal Department of Economic Affairs, Education and Research EAER, Agroscope, CH-3003 Bern, Switzerland; muenger.linda@gmail.com (L.H.M.); guy.vergeres@agroscope.admin.ch (G.V.); 2Max Rubner-Institut, Physiology and Biochemistry of Nutrition, D-76131 Karlsruhe, Germany; ralf.Krueger@mri.bund.de; 3Institute of Microbiology, Lausanne University Hospital and Lausanne University, CH-1011 Lausanne, Switzerland; valentin.scherz@chuv.ch (V.S.); claire.bertelli@chuv.ch (C.B.); gilbert.greub@chuv.ch (G.G.); 4Department of Agricultural and Food Sciences (DISTAL), University of Bologna, I-47521 Cesena, Italy; gianfranco.picone@unibo.it (G.P.); francesco.capozzi@unibo.it (F.C.); 5Service of Endocrinology, Diabetes and Metabolism, Lausanne University Hospital, CH-1011 Lausanne, Switzerland; nathalie.vionnet@chuv.ch

**Keywords:** TMAO, choline, milk, dairy products, fermented milks, microbiota

## Abstract

Trimethylamine-*N*-oxide (TMAO) can be produced by the gut microbiota from dietary substrates and is associated with cardiovascular disease. While dairy products contain TMAO precursors, the effect of fermented dairy on TMAO metabolism remains unclear. We used plasma and urine samples collected for two randomised cross-over studies to evaluate the effects of fermented dairy consumption on TMAO metabolism. In Study 1, thirteen healthy young men tested a yogurt and an acidified milk during postprandial tests and a two-week daily intervention. In Study 2, ten healthy adults tested milk and cheese during postprandial tests. TMAO and five related metabolites were measured in plasma and urine by LC-MS/MS and NMR. Faecal microbiota composition was assessed in Study 1 (16S rRNA metagenomics sequencing). Fermented milk products were associated with lower postprandial TMAO responses than non-fermented milks in urine (Study 1, *p* = 0.01; Study 2, *p* = 0.02) and in plasma, comparing yogurt and acidified milk (Study 1, *p* = 0.04). Daily consumption of dairy products did not differentially affect fasting TMAO metabolites. Significant correlations were observed between microbiota taxa and circulating or urinary TMAO concentrations. Fermentation of dairy products appear, at least transiently, to affect associations between dairy products and circulating TMAO levels.

## 1. Introduction

Trimethylamine-*N*-oxide (TMAO) is an amine oxide that has been associated with the development of cardiovascular disease (CVD) [1,2,3,4,5,6,7,8,9]. Several dietary precursors of TMAO, including betaine, choline and L-carnitine, have also been linked with CVD outcomes [4,5,6]. The gut microbiota, at the interface between diet and internal physiology, plays a critical role in mediating the effects of dietary precursors of TMAO on CVD risk [1,2,3,5]. The metabolism of these molecules to TMAO is dependent on specific microbial metabolism to trimethylamine (TMA) [1,10], an intermediary metabolite which is transported to the liver where it is oxidised to TMAO [11]. Different substrate-dependent synthesis pathways have been described for the microbial production of TMA [12], including choline TMA-lyase enzyme (*cutC*) that acts with the enzyme activator (*cutD*) to metabolise choline [13] and carnitine monooxygenase (*cntA/B*) that metabolises carnitine [14].

The microbial formation of TMA (and consequently TMAO) is conditioned by the availability of dietary TMAO precursors. These molecules are present at high levels in animal products [12]. The comparison of high versus low animal product diets has suggested that diet can influence the formation of TMAO [5,15,16]. Specifically, TMAO was described as one of the most discriminating metabolites that distinguished the urinary metabolomes of lactovegetarians and omnivorous subjects [15], while in a cross-over study that imposed a 15 day vegetarian, low meat or high meat diet, urinary TMAO concentrations were highest after the high meat diet [16]. However, cross-sectional data for an omnivorous cohort of vascular patients suggested that estimations of TMAO dietary precursors were neither associated with circulating TMAO nor carotid plaque burden [17]. Furthermore, the explained variance of TMAO in plasma or urine by current diet and covariables was found to be small in a healthy adult population and appeared to depend upon dietary source of TMAO [18].

Recently, a novel precursor of TMAO that is metabolised by the gut microbiota has been identified in ruminant meat and milk, δ-Valerobetaine [19], contributing to the interest to study the impact of dairy on TMAO metabolism. Other TMAO precursors are found in full fat dairy products, in particular choline, but tend to present at levels lower than other animal products [20,21]. Remarkably, a cross-sectional study found that dietary intake of milk and ‘milk and other dairy foods’ were the only food groups positively associated with plasma TMAO [22]. However, a relative decrease in TMAO was observed in a cross-over study after a short exposure to diets high in either milk or cheese, with respect to the control diet that excluded all dairy produce except butter [23].

The interplay between the host gut microbiota, the bacteria present within fermented dairy products and the variation in TMAO precursors present in dairy products may determine the associations between dairy products and TMAO. In the current investigation, we test the hypothesis that the fermentation of dairy products affects plasma and urinary TMAO concentrations after the consumption of these products. In addition, we sought to explore the potential mechanisms for these effects on TMAO production by assessing metabolites that are implicated in TMAO metabolism and by evaluating the impact of the gut microbiota on dairy-induced TMAO production. Our results suggest that fermented dairy products have different effects to non-fermented dairy products on postprandial TMAO metabolism. The relationship between dairy products and TMAO appears to be influenced by the composition of the gut microbiota.

## 2. Materials and Methods

### 2.1. Study Design and Subjects

Plasma and urine samples were obtained from two randomised, cross-over trials [24,25,26] that assessed the postprandial responses to non-fermented milk and a fermented dairy food. Both studies were conducted at the Centre of Clinical Research, Lausanne (Switzerland), according to the guidelines laid down in the Declaration of Helsinki and were approved by the Ethical Committee of the Canton of Vaud (Switzerland). All study participants gave informed consent for participation. The studies were registered at clinicaltrials.gov: Study 1, NCT02230345; Study 2, NCT02705560.

Full details of the two study designs, populations and test products are published elsewhere [24,25]. Briefly, Study 1 applied a randomised, double-blind, cross-over design to compare postprandial and short-term effects of probiotic yogurt to those of an acidified milk in fourteen healthy young men (Figure 1a) [24]. Subjects were recruited by a poster campaign and screened using strict exclusion criteria, including abnormal clinical biochemistry, the presence of acute or chronic illness, dietary intolerances or allergies, regular medication use and use of antibiotics within the six months prior to the start of the study (Appendix A).

The test foods were a fermented yogurt (containing *Lactobacillus delbrueckii* spp. *bulgaricus*, *Streptococcus thermophilus* and probiotic strain, *Lactobacillus rhamnosus* GG (ATCC 53103- Culture Collection of the University of Goteborg, Sweden, reference CCUG 34291) and a milk acidified with D-(+)-glucono-δ-lactone (2%) to replicate the consistency of yogurt without the addition of a microorganism (nutrient information of the products was previously reported [24]). Both products were prepared from the same batch of milk. The products were evaluated after an overnight fast by a postprandial test comprising a single intake (800 g) of the assigned dairy product (consumed within 15 min), with fasting and 6 h postprandial venous blood and urine sampling. No oral dietary intake except ad libitum water intake was permitted during the test. This marked the beginning of a two-week daily consumption of the assigned dairy product (400 g/d), after which, fasting blood and urine samples were collected. After a three-week wash-out, a second postprandial test and daily consumption period were completed to assess the other dairy product. Prior to each test day, the volunteers followed a three-day controlled diet that was adapted to meet the nutritional requirements of each volunteer. During all study test periods, dietary restrictions limited dairy consumption to products exclusively provided by the study coordinators. A portion of 400 mL/d full fat milk was provided to volunteers for the run-in and wash-out phases.

The second study was a randomised, cross-over study that was completed within the framework of the European JPI-funded project entitled Food Biomarker Alliance (FoodBAll) [27]. As described previously [25], eleven healthy young men (*n* = 6) and women (*n* = 5) were randomly assigned to test three isocaloric test foods: 600 mL milk, 100 g Gruyère cheese (plus 500 mL still water) and 600 mL soya drink (selected as a non-dairy food) (Figure 1b). Subjects were recruited via a poster campaign and screened using exclusion criteria, including the presence of chronic or infectious disease, food intolerances or allergies, anaemia or low ferritin (<30 µg/L) (Appendix A). A pre-test requiring the consumption of 600 mL milk was completed as part of the inclusion protocol to confirm tolerance to milk. On each test day, participants consumed the assigned test food (within 15 min) after a 12 h fast and no further food or drink was permitted during the 6 h postprandial phase except for water (maximum 250 mL/h). Fasting and 6 h postprandial venous blood and urinary sampling were performed to assess the postprandial responses. A fasting plasma sample was also collected 24 h after consumption of the foods. Each test day was preceded by two days ‘run-in’, during which strict dietary restrictions applied that included the total exclusion of dairy and fermented foods, and a standardised meal (chicken, rice and margarine) on the evening prior to the test day that was followed by a 12 h fast. Dietary intake was semi-controlled during the 24 h following consumption of the test food.

### 2.2. Samples

For the purpose of the current analysis, fasting and postprandial, plasma and urine samples were assessed in a targeted manner for selected metabolites (secondary study outcomes). In Study 1, fasting plasma samples taken on each test day (fasting and dairy test days) and nine postprandial plasma samples (0.25, 0.5, 1, 1.5, 2, 3, 4, 5 and 6 h) were collected for each dairy test day. In Study 2, baseline fasting plasma sample (collected before each test) and four postprandial plasma samples (1, 2, 4 and 6 h) collected for the dairy tests (milk and cheese) were assessed in the current analysis. Samples collected on the non-dairy test day (soja) for the primary objective of the study (to assess food biomarkers) are not exploited here as we specifically compare fermented and non-fermented dairy products. All samples were collected and processed according to standard laboratory protocol, before being stored at −80 °C.

Fasting ‘spot’ urine samples were collected for all test days in Study 1 and single pools of urine were collected in a tube with no additives during the postprandial tests (0–6 h). In Study 2, six pooled urine samples (0–1, 1–2, 2–4, 4–6, 6–12 and 12–24 h) were collected in the 24 h following consumption of the test food and were processed as described previously [25]. Urine samples for both studies were stored at −80 °C prior to analysis.

Faecal samples collected in Study 1 during the run-in period and at the end of the wash-out period before the postprandial dairy test day were used for microbiota analysis. In a previous analysis, it was shown that no significant changes in microbiota composition were found between the run-in and wash-out periods [24]. Nevertheless, the samples collected immediately before the postprandial test day under consideration were used here (where available) to evaluate the pre-test microbiota composition. As described previously [24], faecal samples were aliquoted (200 mg) and suspended in 2 mL glycerol-brain heart infusion solution (100 mL glycerol, 37 g brain heart solution, 1 L distilled water) within 4 h of sample collection and under sterile conditions. All samples were stored after homogenisation (agitation 13 × *g*, 10 min) at −80 °C prior to DNA extraction.

One volunteer was previously excluded from Study 1 [24] and one individual who did not participate in the final test day of Study 1 is also not included for any analyses that implied comparisons with this test day (Appendix A). For Study 2, one male volunteer was excluded from the current analysis due to the detection of abnormal fasting glucose values on multiple test days (Appendix A).

### 2.3. Liquid Chromatography–Tandem Mass Spectrometry (LC-MS/MS) Analysis of Trimethylamine-*N*-Oxide (TMAO) and Related Metabolites

Plasma content of TMAO and five related metabolites (carnitine, betaine, choline, N,N-dimethylglycine (DMG) and sarcosine) were assessed for both studies by ultra-performance liquid chromatography–tandem mass spectrometry (UPLC-MS/MS) using an Acquity H-Class coupled with a Xevo TQD (Waters), as described previously [18,28]. Compounds were separated in HILIC mode using an inverse acetonitrile gradient on an Acquity BEH Amide column (1.7 µm, 2.1 mm × 100 mm; Waters) at a flow rate of 0.6 mL/min. Detection was performed by ESI-MRM in positive ion mode with a solvent delay of one minute. The same approach was adapted to quantify the urinary content of these metabolites for Study 1. Method modifications include an extension of the analyte panel (not relevant for the present study), a slightly longer gradient and optimised multiple reaction monitoring time windows for compound quantification. Further details and basic evaluation data concerning the modified method are published elsewhere [29,30]. Samples were diluted (urine, ×25; plasma ×10) with a buffer resembling the LC starting conditions (50% acetonitrile, 50 mM ammonium formate, adjusted to pH 3.2 with formic acid), and 1 µL was injected. Spiked matrix samples were used for calibrators and controls (three levels), and isotope-labelled internal standards were used for calibration. Sample creatinine concentrations were assessed together with the other analytes in the same run and were used to normalise urine metabolite concentrations (division of metabolite concentrations (µm) by creatinine (mg/dL)). Since creatinine concentrations are much higher compared to other compounds, a non-linear calibration and a combination of three internal standards were used.

### 2.4. NMR Assessment of TMAO

TMAO was assessed in urine collected during Study 2 by NMR, as previously described [25]. Briefly, 1H NMR spectra were measured at 298 K using an AVANCETM III HD 600 MHz spectrometer (Bruker BioSpin, Karlsruhe, Germany) (proton frequency 600.13 MHz). After pre-processing steps to acquire spectra, baseline and phase corrections were applied to each spectrum before importing the spectra into Chenomx (v. 7.72) (Chenomx Inc, Edmonton, Canada) with 1 Hz line-broadening for the calculation of TMAO concentration. Absolute quantification of TMAO was achieved using TSP (final concentration of 0.580 mm) as an internal standard. Consequently, no normalisation step was required. For each sample, the concentration of TMAO was established in 540 µL urine that was measured with 60 µL of phosphate buffer solution containing TSP 0.1%. The quantity of TMAO in the original urine sample was thus derived by correcting for the addition of the internal standard (division by 0.9) and multiplying by the total volume of the original sample.

### 2.5. 16S rRNA Metagenomic Analysis

As described previously [24], faecal DNA was extracted with QIAamp Fast DNA Stool Mini Kit (Qiagen, Hilden, Germany). DNA libraries prepared according to the “16S Metagenomic Sequencing Library Preparation” (Part. # 15044223 Rev. B) protocol from Illumina^®^ (San Diego, CA, USA) were sequenced on Illumina^®^ MiSeq. Raw reads were processed using the DADA2 R package (1.8.0) [31], integrated in a Snakemake (5.3.0) [32] pipeline to identify amplicon sequence variants (ASV), which confer a higher resolution than traditional operational taxonomic unit assignment. Taxonomy was assigned using the Ribosomal Database Project classifier [33] (QIIME 1.9.1 wrapper [34]) against the EzBioCloud database (2017-01) [35] and integrated with ASV counts in Phyloseq (1.24.2) [36]. Taxonomic classification of features displaying significant association (see below, Statistical Analysis) was verified against the Genome Taxonomy Database (GTDB) [37] (GTDB 16S, revision 89-modified), using the online Decipher classifier [38] (accessed on 2019/09/23, 50% confidence threshold). ASV counts were used for functional metagenomic prediction using the PICRUSt2 pipeline (2.1.0_b) [39,40,41,42,43]. PICRUSt2 assigns ASV to reference genomes from which functional and protein annotation were pre-extracted to depict putative proteins and functions contained within a metagenome. The gene encoding the choline TMA-lyase enzyme that converts TMA from choline was estimated by targeting the TIGR04394 protein family (*cutC*) [44] of the TIGRFAM annotation.

### 2.6. Statistical Analysis

All statistical analyses were carried out in the R environment (version 3.5.3) [45]. Baseline inter-day variation in the measured TMAO metabolites was assessed by comparing values of all TMAO metabolites in fasting samples of plasma and urine between the two dairy test days. The postprandial changes of circulating metabolites in response to the dairy tests were evaluated by comparison of the delta change (with respect to fasting levels) for each time point and by comparison of the net incremental area under the curve (iAUC) for the 6 h postprandial period (linear trapezium method). The postprandial changes in urinary TMAO metabolites was evaluated for Study 1 by comparing the metabolite concentrations in the 6 h pooled urine samples between the two dairy tests, while for Study 2, pairwise comparisons were completed for the cumulative quantities of TMAO at each time interval assessed during the 24 h following each test. Cumulative quantities of urinary TMAO were calculated by summation of TMAO found in all the pools within the time interval being assessed, minus TMAO found in baseline (fasting) samples. For Study 1, the effects of a two-week daily consumption of dairy on plasma and urine metabolites was evaluated by first assessing for each dairy product whether a significant change in fasting levels was observed pre versus post the two-week test period. Subsequently, fasting values after the two-week daily consumption of yogurt versus acidified milk were compared. For all analyses, the measures for the two dairy tests were compared using a paired Wilcoxon signed rank test (statistical significance considered at *p* < 0.05). Carryover effects were verified where significant differences were observed to confirm the assumption of negligible carryover effects, according to the procedure described by Wellek and Blettner [46].

Correlation analyses assessed the relationship between TMAO metabolites in blood and urine using Spearman’s correlation test. Associations for each postprandial and fasting condition, sample type (i.e., blood or urine) and for each study test day were assessed separately (significance threshold of false discovery rate (FDR) < 0.05 [47] to account for multiple testing). Where significant differences in fasting or postprandial responses of metabolites were observed between fermented and non-fermented dairy products in both urine and plasma, correlation analyses were conducted to confirm that the trends were associated.

A post hoc microbiota analysis sought to investigate whether the dynamic postprandial response in TMAO observed in both plasma and urine after the acidified milk test could be explained by features of the pre-test microbiota composition. Associations between relative abundance of the microbiota taxa and TMAO (assessed in fasting (baseline) plasma and in urine and plasma collected in the 6 h postprandial phase after acidified milk consumption) were assessed using the ALDEx2 package [48,49,50]. The same analysis strategy was used to evaluate the association between the estimated abundance of predicted *CutC* and TMAO concentrations in plasma and urine. ALDEx2 was used to convert counts to centred log-ratio transformed values (using Monte Carlo sampling), before assessing associations between these transformed counts and TMAO concentrations in plasma and urine using Spearman’s rank correlation. Correlations were assessed for taxa (detected in at least five samples) with counts grouped by hierarchical assignment for all taxonomic levels from ASV to class. Significant associations were considered if FDR < 0.10 to account for multiple testing [47]. For the targeted assessment of the protein TIGR04394, significant associations were considered where *p* < 0.05.

## 3. Results

### 3.1. Subject Baseline Characteristics and Samples

The baseline subject characteristics for Studies 1 and 2 confirmed that all subjects were young (aged 18–39 y), healthy adults (median BMI respectively 22.09 kg/m^2^ (IQR 20.0–22.69) for Study 1 and 21.8 kg/m^2^ (IQR 20.8–23.3) for Study 2), as reported previously [24,25]. Targeted metabolite analysis was completed for samples from thirteen subjects in Study 1 and ten subjects of Study 2. One fasting urine sample was not available for Study 1 (sample after two-week dairy product consumption) while two urine samples were missing for Study 2 (6–12 h milk and 12–24 h milk). In both cases, the corresponding data points for the other dairy test were removed in view of the paired testing approach. Baseline fasting plasma and urine metabolite concentrations were not significantly different between the dairy test days for either study (*p* > 0.05).

### 3.2. Postprandial Changes in TMAO after Dairy Consumption

The 6 h postprandial TMAO response (net iAUC) in plasma was significantly lower after yogurt compared to acidified milk consumption in Study 1 (*p* = 0.04) (Figure 2a). Lower concentrations of TMAO were found after yogurt consumption for the comparisons at each time point, with significant differences at 3 h (*p* = 0.03) (Appendix A). In accordance with the differences observed for the plasma postprandial responses to acidified milk and yogurt, postprandial urinary TMAO concentrations in the 6 h postprandial phase were significantly lower after yogurt than acidified milk (*p* = 0.01, Figure 2b). These results were partially replicated in Study 2 where a significantly lower cumulative TMAO urinary excretion was observed after cheese consumption compared to milk for the interval 0–6 h (*p* = 0.02, Figure 2d), while plasma TMAO levels were not significantly lower after cheese compared to milk consumption (net iAUC, *p* = 0.10) (Figure 2c, Appendix A).

The biological variation of plasma and urine TMAO was consistently lower after fermented dairy compared to non-fermented milk consumption (Appendix A). The similar trends in the postprandial changes in TMAO in plasma and urine were significantly positively associated in the correlation analyses for Study 1 (Figure 3a) but associations were not significant in Study 2 (Figure 3b).

### 3.3. Postprandial Changes in TMAO-Related Metabolites and Dairy Consumption

Of the five TMAO-related metabolites that were assessed, no significant differences in the 6 h postprandial responses were found comparing yogurt and acidified milk (Appendix A). Conversely, significantly lower postprandial responses were observed for betaine and choline after cheese compared to milk consumption in Study 2 (Appendix A). Different effects of the dairy products on metabolite concentrations at specific postprandial time points were observed for DMG and sarcosine (Appendix A). DMG increased significantly more at 2 h after yogurt compared to acidified milk consumption (*p* = 0.002, Appendix A), while a relative decrease in DMG, which was significantly lower after cheese consumption compared to that of milk, was observed in Study 2 at 4 h (*p* = 0.03) and 6 h (*p* = 0.02) (Appendix A). No postprandial differences for sarcosine were observed comparing yogurt and acidified milk (Appendix A), though higher postprandial levels of the metabolite were observed at 2 h after cheese consumption compared to that of milk (*p* = 0.03, Appendix A). No differences in carnitine postprandial responses to fermented compared to non-fermented milk consumption were observed for either study (Appendix A).

In Study 1, the 6 h urinary excretion of DMG and betaine were also significantly different comparing yogurt and acidified milk (Appendix A). The significantly higher concentrations of urinary DMG after yogurt as compared with acidified milk corresponded to the differences of the metabolite in plasma at 2 h. Conversely, while the postprandial changes in plasma betaine were not different after yogurt compared with acidified milk, the 6 h urinary excretion of betaine was significantly greater after yogurt consumption. These metabolites were not evaluated in urine for Study 2.

### 3.4. Sustained Effects of Dairy Consumption on TMAO and Related Metabolites

A repeated daily exposure to the dairy products did not have a marked effect on TMAO or TMAO-related metabolites in plasma and urine in Study 1. Fasting plasma levels in TMAO were not significantly different after two-week daily consumption of acidified milk compared to yogurt, and neither intervention caused a significant change in fasting plasma with respect to fasting concentrations taken before the intervention (Appendix A). The differences observed in the postprandial urinary excretion of TMAO after a single 800 g dose of yogurt compared with acidified milk were not found in the fasting urinary samples that were taken after two-week daily consumption of the dairy products (*p* = 0. 23). In Study 2, no significant effects of the dairy products on TMAO metabolism beyond the 6 h postprandial phase were found in the 12 and 24 h assessments.

### 3.5. Correlations between TMAO and Related Metabolites

#### 3.5.1. Fasting Analyses

Correlation analyses for fasting plasma metabolites revealed a positive association between sarcosine and DMG. This was found in three of four fasting assessments in Study 1 (rho = 0.66, 0.56 and 0.43 respectively, for all fasting samples except after two-week acidified milk consumption, FDR < 0.05). Positive correlations for the same metabolites measured in fasting conditions were also observed in Study 2 but these associations were not significant. Urinary choline was positively associated with urinary TMAO in Study 1, but this association was only significant for the assessment before acidified milk (rho = 0.42, FDR = 0.02).

#### 3.5.2. Postprandial Analyses

Similar postprandial responses were observed for carnitine, betaine and choline in Study 1 (Figure 4). For both acidified milk and yogurt, positive associations were observed in pairwise comparisons of the three metabolites. These associations were not reproduced in Study 2. Interestingly, the correlations between the same metabolites in 6 h urine samples only showed a significant association between choline and betaine, specifically after acidified milk consumption (rho = 0.35, FDR = 0.03). Other correlations indicated that the circulating postprandial response in TMAO after acidified milk consumption was inversely associated with that of sarcosine in Study 1 (rho = −0.52, FDR = 0.03) but not significantly after milk in Study 2 (rho = −0.30, FDR = 0.42). Additionally, the TMAO response to cheese was positively associated with that of carnitine (rho = 0.59, FDR = 0.02) in Study 2.

### 3.6. Correlations between Microbiota Taxa and TMAO Concentrations in Blood and Urine

Several inverse associations were identified between fasting TMAO concentrations in plasma and microbiota taxa (Table 1). These associations included two genera from the Clostridiales order: *Eisenbergiella*, from the *Lachnospiraceae* family and one putative genus (*EU844456_g*) placed *incertae sedis* within the Clostridiales order and the *Mogibacterium_f* family (a phylotype introduced by the EzBioCloud database authors [35]). In classification confirmation using GTFB, this feature corresponded to an undefined genus of the proposed *Anaerovoracaceae* family. This family was also significantly inversely associated with fasting TMAO concentrations in plasma.

A significant positive association was observed between an uncharacterised genus of the *Lachnospiraceae* family, *AY305316_g* (*CAG-81* in GTDB), and the 6 h postprandial urinary TMAO concentrations after acidified milk consumption (Figure 5). An inverse trend was also observed between 6 h postprandial urinary TMAO concentrations after acidified milk consumption and the class Lentisphaeria (rho = −0.65, *p* = 0.03, FDR = 0.19). Conversely, no significant associations were observed between postprandial circulating TMAO after acidified milk and microbiota taxa. No associations were observed between TMAO plasma or urinary concentrations in fasting or postprandial conditions and the pooled abundance of all taxa predicted to express the *cutC* gene (*p* > 0.05).

## 4. Discussion

In two human diet intervention studies, we showed lower circulating and urinary TMAO in the postprandial phase after fermented dairy (yogurt and cheese) consumption compared to non-fermented dairy consumption. These results were more marked in urine though we found correlations between plasma and urine TMAO after acidified milk and yogurt interventions. In addition, we identified some associations between TMAO concentrations (blood and urine) and taxa of the gut microbiota.

Evidence for a modulatory role of dairy products on TMAO concentrations in blood or urine is inconsistent [18,22,23,51] but this may be explained in part by the type of dairy product considered. Indeed, Rohrmann et al. [22] already showed that while ‘milk and dairy products’ were positively associated with plasma TMAO, when ‘milk consumption’ and ‘cheese consumption’ were assessed separately, the association was only present for milk consumption. Conversely, we previously found no association between ‘milk and dairy products’ and spot urine TMAO concentrations, but highlighted a trend for an inverse association between ‘cheese and curd cheese’ consumption and spot urine TMAO [18]. Certainly, to consider dairy foods as a single, uniform food group does not account for the highly variable macronutrient content, TMAO substrate availability or presence of bacteria in different dairy products, all of which can influence TMAO metabolism [1,51,52,53,54,55].

Our observations of lower postprandial TMAO after fermented dairy intake were evident both when dairy products were matched for caloric load and fat content (Study 1) [24], as well as products only matched for calorie load but not for macronutrients (Study 2) [25]. It was noteworthy that different biological variances in postprandial TMAO responses were observed, depending on whether fermented or non-fermented dairy was consumed. The comparably high inter-individual variation in the case of the non-fermented milks points to the known influence of various parameters on TMAO metabolism, including gut microbiota, enzymatic activity and substrate bioavailability. This usual, well-known variability may be reduced by consumption of bacteria-containing probiotic food such as yogurt and cheese. This would support a certain levelling of TMAO production as one major aspect of fermented versus non-fermented dairy.

The postprandial TMAO differences that we observed did not extend to a sustained effect in the 24 h after dairy consumption (Study 2) or following the two-week daily intervention (Study 1). These results suggest that the observed postprandial increase after non-fermented dairy remains within a physiological dynamic range that can rapidly stabilise to baseline levels. This is also supported by the relatively early peak difference in plasma TMAO at 3 h (Study 1) and by the rapid excretion of the metabolite that was confirmed in our urinary data. While some circulating TMAO can accumulate in tissues, it is principally excreted by the kidneys [56,57]. Our findings are in agreement with a cross-over study that could not confirm a difference in 24 h urinary excretion of TMAO after 14 days of milk- or cheese-enriched diets, though both dairy products in this case were associated with significant reductions in TMAO relative to a low-dairy control diet [23].

We found close associations between the three dietary precursors of TMAO, betaine, carnitine and choline during the postprandial phase, but these metabolites did not show large differences in responses when comparing our fermented and non-fermented dairy products. This seems to correspond with the lack of associations between dairy consumption and these metabolites reported by Rohrmann et al. [22]. All of the three precursors are present in dairy products, but for the foods we tested, we estimated that total choline (14–16 mg/100 g) [20] was much higher than betaine (0.6–0.8 mg/100 g) [20] or L-carnitine content (1.4–4.2 mg/100 g) [21]. Estimates for the choline content per portion of the dairy products used in Study 1 were identical (112 mg), corresponding to the lack of difference in the postprandial dynamics of this metabolite. However, although eucaloric, the cheese and milk portions in Study 2 were not matched for total choline content: estimated 16 mg cheese, 84 mg milk. These choline differences could be an indirect cause of the higher plasma betaine observed after milk compared to cheese consumption, though plasma choline actually decreased after both dairy products. At first glance, the close association between betaine, carnitine and choline in the postprandial phase but not under fasting conditions is suggestive of a common postprandial dynamic rather than confirmatory of a true relationship. However, given that the metabolism of choline in the liver and kidney results in its major metabolite betaine, which can be used to synthesise carnitine [58], the endogenous production of betaine and carnitine could explain the observed associations.

A role of the gut microbiota on the postprandial increase in TMAO after acidified milk consumption was suggested by the positive association between an uncharacterised genus of the *Lachnospiraceae* family that comprises genera known to express the *cutC* gene, [9,59] and 6 h postprandial urinary TMAO concentrations after acidified milk consumption in Study 1. While this association should be interpreted with some caution given the limited number of subjects included in this analysis, it was interesting to note that the *Lachnospiraceae* family have previously been positively associated with TMAO and TMA in a mice model, with a trend for an association with an increased aortic plaque area [9]. Conversely, a different genus of the *Lachnospiraceae* family (*Eisenbergiella*) was inversely associated with fasting plasma TMAO levels. The finding that two different genera of the same *Lachnospiraceae* family exhibit completely different effects on TMAO levels in our study is in line with previous reporting of wide but uneven distribution of *cutC* at the genus level and among four different bacterial phyla [60]. This underlines the importance of using adapted amplicon-based metagenomics approaches (in terms of gene-targeted, amplicon length and reference databases) or shotgun metagenomics in the future to allow discrimination of bacteria below the genus level. Our targeted assessment of bacteria with the *cutC* gene did not confirm an association with these bacteria and TMAO levels after acidified milk consumption. This may reflect the limitations of the predictive bioinformatic approach that we applied which, despite the use of ASV, could not capture fine, strain-level presence or absence of the genes of interest or the variation of gene expression. The lack of associations could also in part be due to the small sample size in our study, considering that thousands of taxa and their in silico inferred genes are assessed.

The inverse associations that were identified between the microbiota and baseline fasting TMAO levels evoke the notion of a competitive influence of the microbiota community or individual bacteria on TMA-producing bacteria, which has even been suggested as a therapeutic strategy for TMAO-associated CVD pathologies [9]. Indeed, there has been interest to use probiotic strains to reduce plasma TMAO elevated, for example, due to a high-fat diet [54] or choline-rich diet [55]. As many fermented dairy products, including those used in the current study, contain probiotic bacteria, the dietary effects of fermented dairy products on TMAO metabolism imply a complex interaction of the bacteria present in the product itself, the quantity of TMA dietary substrates and the microbiota composition of the consumer.

A strength in the current work is the use of two independent studies. This allowed us to assess the robustness of our findings on postprandial TMAO dynamics in different dairy products. It was also useful to compare the urinary metabolite changes to those observed in plasma to verify the metabolic fate of the biomarker as well as to confirm that the urinary dynamics corresponded to those of plasma. The validity of the results was supported by the consistent findings for TMAO in urine for Studies 1 and 2, despite using two different approaches, NMR and mass spectrometry. Furthermore, the use of cross-over study designs with dietary restrictions during the wash-out phases between each test were chosen to control for inter-individual variation of TMA-producing strains of the microbiota that may also account for some of the differences in the reported effects of dairy products on TMAO.

The effects of dairy food consumption on TMAO production seems to depend on the type of dairy product consumed and the presence of TMA-producing bacteria in the gut microbiota. In the postprandial phase, we observed differences between fermented and unfermented dairy foods on urinary and circulating TMAO concentrations. However, further investigations are required to understand the specificity of these observations to fermentation and to confirm potential health consequences of such differences.

## Figures and Tables

**Figure 1 nutrients-12-00234-f001:**
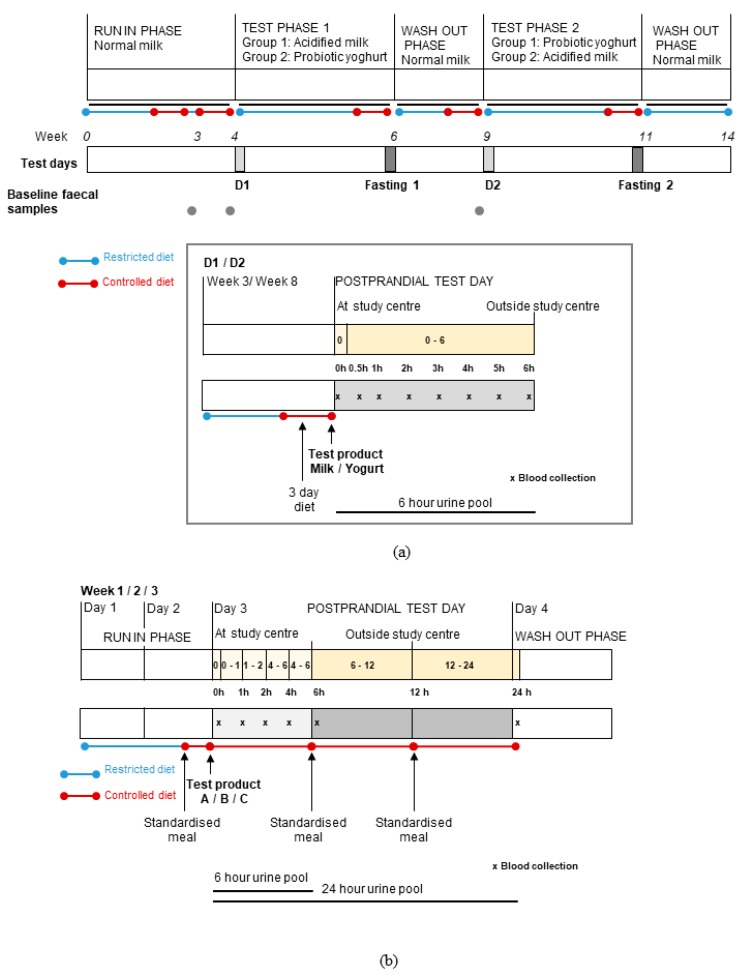
Overview of the study designs used for two randomised, controlled, cross-over trials that compared the postprandial and short-term effects of consuming fermented or non-fermented dairy products (adapted from Burton et al. and Münger et al.) [24,25]. (**a**) In Study 1, volunteers (*n* = 14) were assigned to test probiotic yogurt and acidified milk in random order during two test phases. At the beginning of each test phase, postprandial dairy tests (D1 and D2) were conducted while the effects of two-week daily consumption of each product were assessed by fasting tests (Fasting 1 and 2). (**b**) In Study 2, test products were administered to volunteers (*n* = 11) in random order (A = milk, B = cheese, C = Soy drink) and their effects were evaluated by postprandial tests. For both studies, on postprandial test days, blood (**x**) and urine sampling (intervals shown by yellow blocks) was completed as indicated on the schematics. Fasting blood and urine samples were collected on each test day. In Study 1, pre-test faecal samples were collected during the run-in and wash-out phases. Dietary restrictions were applied during the studies with the provision of a three-day controlled diet (Study 1) and a standardised meal (Study 2) before each test day.

**Figure 2 nutrients-12-00234-f002:**
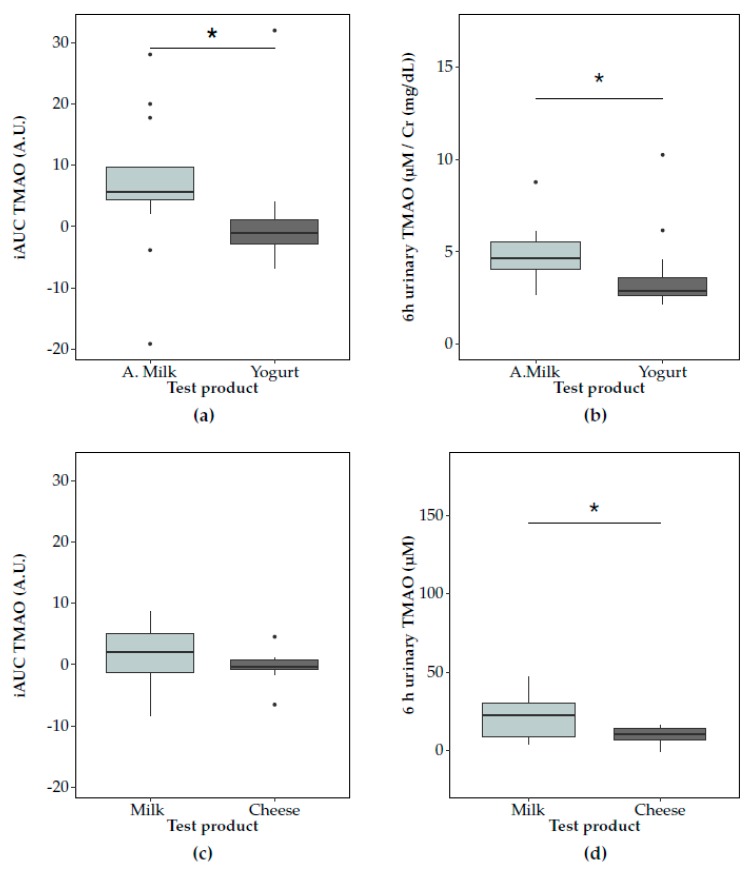
Postprandial trimethylamine-*N*-oxide (TMAO) responses in plasma (summarised by net incremental area under the curve (iAUC)) and urine after consumption of non-fermented milk (light grey) and a fermented milk product (dark grey) consumption. In Study 1, TMAO responses to acidified milk (‘A.milk’) and yogurt are compared in (**a**) plasma and (**b**) urine. In Study 2, TMAO responses to milk and cheese are compared in (**c**) plasma and (**d**) urine. The concentrations of TMAO after the dairy products’ tests are compared using the paired Wilcoxon signed-rank test (significance at *p* < 0.05, indicated by *). Plots show the IQR (box), the median dividing the IQR (―), with whiskers that extend to the last point included in the 1.5 × IQR range and outliers outside this range identified (•).

**Figure 3 nutrients-12-00234-f003:**
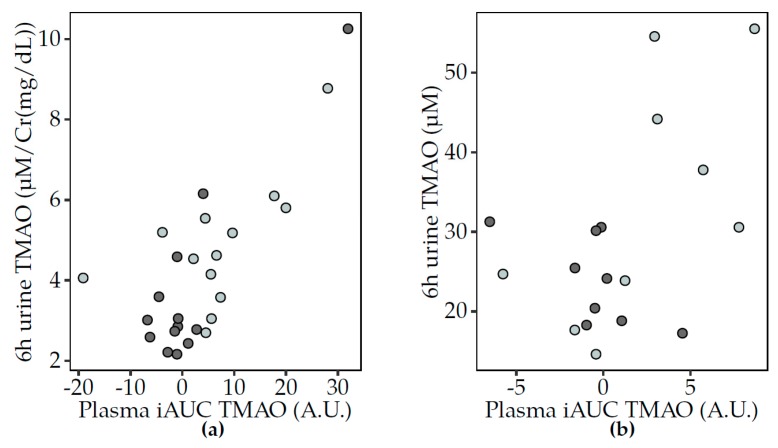
Scatter plots to illustrate the association between the postprandial TMAO changes in plasma (iAUC) and urine (6 h pool) after non-fermented milk (light grey) and fermented dairy (dark grey) consumption. Spearman’s correlation test showed positive associations between postprandial TMAO (iAUC) in plasma and urine (6 h pool) in Study 1 (**a**) (acidified milk, rho = 0.48, *p* = 0.02; yogurt, rho = 0.39, *p* = 5 × 10^−5^) and after milk consumption in Study 2 (**b**) (milk, rho = 0.52, *p* = 0.15; cheese, rho = 0.11, NS). Significant associations at *p* < 0.05. iAUC, incremental area under the curve; TMAO, trimethylamine-*N*-oxide; Cr, creatinine.

**Figure 4 nutrients-12-00234-f004:**
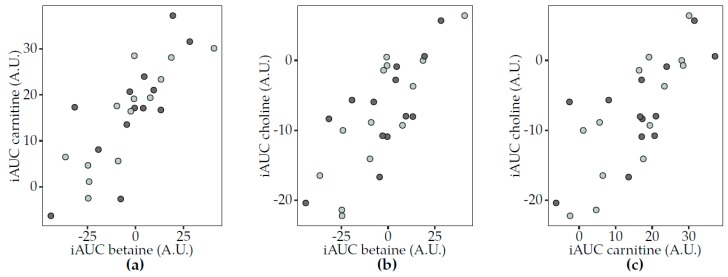
Scatter plots to illustrate associations between the postprandial responses (iAUC) in plasma of selected circulating metabolites to acidified milk (light grey) and yogurt (dark grey) consumption (Study 1). Spearman’s correlation test completed for acidified milk and yogurt (significance for FDR < 0.05) showed positive correlations for (**a**) carnitine and betaine (acidified milk, rho = 0.87, FDR = 0.003; yogurt, rho = 0.74, FDR = 0.03), (**b**) choline and betaine (acidified milk, rho = 0.84, FDR = 0.003; yogurt, rho = 0.63, FDR = 0.05) and (**c**) choline and carnitine (acidified milk, rho = 0.80, FDR = 0.003; yogurt, rho = 0.58, FDR = 0.06). A.U., arbitrary units; FDR, false discovery rate; iAUC, incremental area under the curve.

**Figure 5 nutrients-12-00234-f005:**
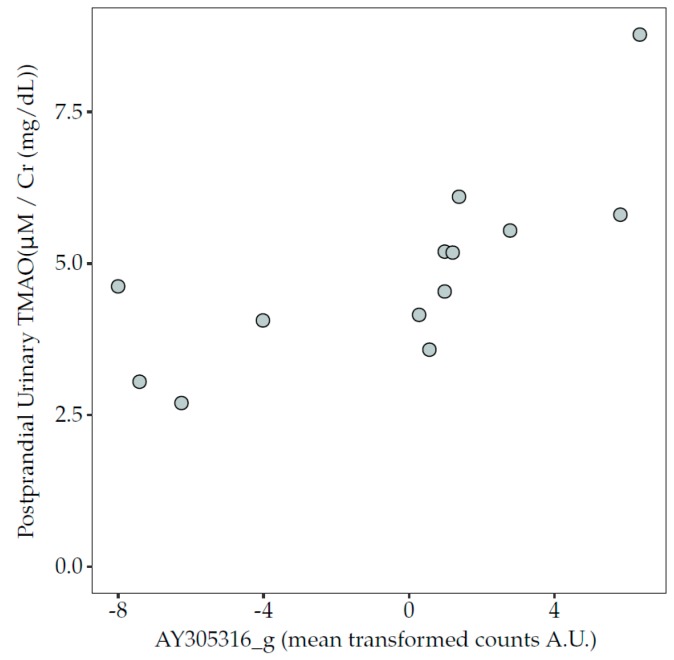
Association between the relative abundance of genus *AY305316_g* (centred, log-ratio transformed counts) and concentration of TMAO in 6 h pooled urine collected in the postprandial phase after the acidified milk test (Spearman’s correlation, rho = 0.84, *p* = 0.0008, FDR = 0.06). FDR, false discovery rate; TMAO, trimethylamine-N-oxide.

**Table 1 nutrients-12-00234-t001:** Spearman’s correlation between fasting plasma TMAO and relative abundance of microbiota taxa.

**Family (EzBioCloud)**	**Family (GTDB)**	**Rho**	***p* Value**	**FDR**
*Mogibacterium_f*	*Anaerovoracaceae*	−0.61	0.006	0.07 *
*Christensenellaceae*	Unclassified Bacteria	−0.58	0.006	0.08 *
**Genus (EzBioCloud)**	**Genus (GTDB)**	**Rho**	***p* Value**	**FDR**
*Eisenbergiella*	*Eisenbergiella*	−0.67	0.001	0.05 *
*EU844456_g*	Unclassified Anaerovoracaceae	−0.63	0.003	0.09 *

* Significance for associations with FDR < 0.10. FDR, false discovery rate; TMAO, trimethylamine-*N*-oxide.

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
