# Peer review of "Trimethylamine-N-Oxide Postprandial Response in Plasma and Urine Is Lower After Fermented Compared to Non-Fermented Dairy Consumption in Healthy Adults"

_nutrients, 2020, doi:10.3390/nu12010234_

Round 1
Reviewer 1 Report
The authors have demonstrated that the fermented dairy products (probiotic yoghurt or cheese) have an effect on the TMAO concentrations in blood and urine decreasing the concentration of this compound (associated with the rsik of cardiovascular dieseases) in comparison to either acidified milk or ordinary milk. There was also an indication of intestinal microbiota changes that were assosciated with the amount of circulating TMAO.
The study appears sound and the results reliable. However, the number of test subjects was relatively limited (14 in the study, where the faecal microbiota was evaluated), and I woud be cautious to draw conclusions on the associations of specific intestinal microbial taxa and the TMAO formation from this small number of test subjects. I hope that the authors would comment on this in their discussion.
Some specific questions:
1) "dairy tests" on line 98, possibly a typo?
2) There appears to be a duplication of information on lines 124 - 126 and 129 - 131.
3) Omission of the analysis of the effects of the soya product (line 147) in the study 2 ahould be justified.
Author Response
Response to Reviewer 1 Comments
The authors have demonstrated that the fermented dairy products (probiotic yoghurt or cheese) have an effect on the TMAO concentrations in blood and urine decreasing the concentration of this compound (associated with the rsik of cardiovascular dieseases) in comparison to either acidified milk or ordinary milk. There was also an indication of intestinal microbiota changes that were assosciated with the amount of circulating TMAO.
The study appears sound and the results reliable. However, the number of test subjects was relatively limited (14 in the study, where the faecal microbiota was evaluated), and I woud be cautious to draw conclusions on the associations of specific intestinal microbial taxa and the TMAO formation from this small number of test subjects. I hope that the authors would comment on this in their discussion.
We agree with the reviewer that the relatively small number of subjects assessed in our study does limit the strength of the associations with the intestinal microbiota. We had previously referred to this in the following lines of the discussion:
‘The lack of associations could also in part be due to the small sample size in our study considering that thousands of taxa and their in silico inferred genes are assessed.’
However, we have now added a comment in lines 453-455 to highlight this weakness after the description of the clearest association between TMAO formation and specific intestinal microbiobial taxa:
‘While this association should be interpreted with some caution given the limited number of subjects included in this analysis, it was interestingly to note that the Lachnospiraceae family have previously been positively associated with TMAO and TMA in a mice model, with trend for an association with increased aortic plaque area [9].’
Some specific questions:
1) "dairy tests" on line 98, possibly a typo?
Throughout the text, the postprandial tests using cheese, milk or yogurt are referred to as ‘dairy tests’. The description in line 98 is consistent with this.
2) There appears to be a duplication of information on lines 124 - 126 and 129 - 131.
Thank you for highlighting this error that is now corrected to retain the description in the first instance.
3) Omission of the analysis of the effects of the soya product (line 147) in the study 2 ahould be justified.
We have added an explanation of why the response to the soya product was not evaluated in lines 148-150:
‘Samples collected on the non-dairy test day (soja) for the primary objective of the study (to assess food biomarkers) are not exploited here as we specifically compare fermented and non-fermented dairy products’.
Reviewer 2 Report
This is a good, nicely performed and well written paper on an albeit important aspect on the role of fermented dairy on TMAO metabolism. I have some minor comments for the authors consideration.
Lines 48-49: But TMA did not oxidize TMAO in the liver? ,,, and it was the microbiota that previously formed TMA ... In this sentence indicate that both are formed in by the microbiota ..
Line 87: in the Figure 1, the authors indicate 13 participants
Line 124: In the figure 1 the authors indicate 10 volunteers
Line 163: this explain the differences in the count of participants (lines 87 and 124). However it is important to homogenize the number in the different parts within material and method section (text and figures)
Lines 187-188: this sentence is repeated with that of lines 177-178
Line 191: In this NMR analysis of TMAO, is it not necessary the normalization of the results with the creatinine contents?
Author Response
Response to Reviewer 2 Comments
This is a good, nicely performed and well written paper on an albeit important aspect on the role of fermented dairy on TMAO metabolism. I have some minor comments for the authors consideration.
Lines 48-49: But TMA did not oxidize TMAO in the liver? ,,, and it was the microbiota that previously formed TMA ... In this sentence indicate that both are formed in by the microbiota ..
The authors agree that the current phrasing is ambiguous. TMAO is indeed formed in the liver but the production of TMA is dependant upon the microbiota production of TMA from TMAO dietary substrates.
We have revised to help improve clarity in line 48:
‘The microbial formation of TMA (and consequently TMAO) is conditioned by the availability of dietary TMAO precursors’.
Line 87: in the Figure 1, the authors indicate 13 participants
Line 124: In the figure 1 the authors indicate 10 volunteers
Line 163: this explain the differences in the count of participants (lines 87 and 124). However it is important to homogenize the number in the different parts within material and method section (text and figures)
To respond to these three points, both figures 1 A and B, we had indicated only the number of subjects included in final analysis. We have amended the figures as suggested by the reviewer. The explanation of the final number of subjects included in the analysis is kept in the material and methods text. We note that in the abstract the total number of subjects is the number included in the final analysis to give the readers a proper understanding of size of the sample used to obtain the results.
Lines 187-188: this sentence is repeated with that of lines 177-178
This repetition arose from a previous revision that provided a shorter description of the method. We have now corrected and restructured the paragraph (lines 172-199) to a former version of the manuscript.
Line 191: In this NMR analysis of TMAO, is it not necessary the normalization of the results with the creatinine contents?
In the NMR analysis, correction for creatinine is not required as we assume that clearance for the same subject is constant during the excretion of the 6 pooled urine samples taken over 24 hours. Moreover, we measured the cumulative amount of TMAO, starting with an increase from baseline and ending with a return to the baseline concentration, which gives us a correct estimate of TMAO quantity associated to food intake.
Reviewer 3 Report
The manuscript entitled "Trimethylamine-N-Oxide Postprandial Response in Plasma and Urine is Lower After Fermented Compared to Non-Fermented Dairy Consumption in Healthy Adults" is well written and will be of interest to potential readers. Some minor comments are provided below:
L50-51: It might be useful to give examples of the high and low animal product diets referred to in the cited article
THe authors refer to "acidified millk", was there any microbe added apart from the GDL?
Author Response
Response to Reviewer 3 Comments
The manuscript entitled "Trimethylamine-N-Oxide Postprandial Response in Plasma and Urine is Lower After Fermented Compared to Non-Fermented Dairy Consumption in Healthy Adults" is well written and will be of interest to potential readers. Some minor comments are provided below:
L50-51: It might be useful to give examples of the high and low animal product diets referred to in the cited article
In lines 50-51, we cite several articles but in the next lines (51-54), we specify the nature of the diets in two of the three cited articles in more detail (lactovegetarians versus omnivorous subjects, vegetarian versus low meat or high meat diets). We have now clarified this connection by using a different linking word in the second sentence.
The authors refer to "acidified millk", was there any microbe added apart from the GDL?
There is no microbe added to acidify the milk. This is now clarified in line 110:
‘….and a milk acidified with D-(+)-glucono-δ-lactone (2%) to replicate the consistency of yogurt without the addition of a microorganism (nutrient information of the products was previously reported [24]).’